# SARS-CoV-2 Vaccine-Induced Immune Thrombotic Thrombocytopenia with Venous Thrombosis, Pulmonary Embolism, and Adrenal Haemorrhage: A Case Report with Literature Review

**DOI:** 10.3390/vaccines10040595

**Published:** 2022-04-12

**Authors:** Hauke Christian Tews, Sarah M. Driendl, Melanie Kandulski, Christa Buechler, Peter Heiss, Petra Stöckert, Klaus Heissner, Michael G. Paulus, Claudia Kunst, Martina Müller, Stephan Schmid

**Affiliations:** 1Department of Internal Medicine I, Gastroenterology, Hepatology, Endocrinology, Rheumatology and Infectious Diseases, University Hospital Regensburg, Franz-Josef-Strauss-Allee 11, 93053 Regensburg, Germany; sarah.driendl@ukr.de (S.M.D.); melanie.kandulski@ukr.de (M.K.); christa.buechler@ukr.de (C.B.); petra.stoeckert@ukr.de (P.S.); klaus.heissner@ukr.de (K.H.); claudia.kunst@ukr.de (C.K.); martina.mueller-schilling@ukr.de (M.M.); stephan.schmid@ukr.de (S.S.); 2Department of Radiology, University Hospital Regensburg, University Hospital Regensburg, Franz-Josef-Strauss-Allee 11, 93053 Regensburg, Germany; peter.heiss@ukr.de; 3Department of Internal Medicine II, University Hospital Regensburg, Franz-Josef-Strauss-Allee 11, 93053 Regensburg, Germany; michael.paulus@ukr.de

**Keywords:** SARS-CoV-2, COVID-19, vaccine, thrombocytopenia, VITT, adrenal insufficiency

## Abstract

Vaccine-induced immune thrombotic thrombocytopenia (VITT) with venous thrombosis is a rare complication of SARS-CoV-2 vaccination with ChAdOx1 (AstraZeneca) and AD26.COV2.S (Johnson & Johnson, New Brunswick, NJ, USA) associated with high mortality. At present, there are no known differences in the pathophysiology or risk factors of VITT with the AstraZeneca vaccine (ChAdOx1) compared with the Johnson & Johnson vaccine (AD26.COV2.S). Herein, we present the case of a healthy 39-year-old patient with VITT after having received the vaccine Ad26.COV2.S. Ten days after vaccination, the patient developed a deep vein thrombosis and subsequent pulmonary embolism. A computed tomography scan of the abdomen showed adrenal gland bleeding and an adrenocorticotrophic hormone stimulation test diagnosed adrenal insufficiency. Therapy with intravenous immunoglobulin, argatroban and hydrocortisone was initiated immediately after diagnosis. The patient left the hospital 22 days after admission with the diagnosis of adrenal insufficiency but otherwise in good health. To the best of our knowledge, five cases of VITT and adrenal bleeding have been described to date in the literature but the presented case was the first to occur after immunisation with the vaccine of Johnson & Johnson. In summary, VITT-associated adrenal dysfunction is a very rare complication of vaccination with an adenoviral vector-based COVID-19 vaccine.

## 1. Introduction

The COVID-19 pandemic has to date resulted in approximately 472 million confirmed cases and at least 6.1 million deaths. According to the World Health Organization, there are approximately 1.9 million new cases every day [1]. Globally, approximately 11 billion vaccine doses have been administered [1]. Vaccination protects against severe illness, reduces the probability of getting infected, and is the most promising approach to end the pandemic. Highly effective vaccines were developed very quickly and have been authorised for use by the U.S. Food & Drug Administration (FDA) and/or European Medicines Agency (EMA) [2,3,4]. Their safety and efficacy were approved after thorough evaluation, which is required for any new vaccine in Europe and the United States. The most common adverse events were pain at the site of injection, myalgia, headache, nausea, and pyrexia. Post-marketing adverse events for the AstraZeneca vaccine were rare cases of thrombosis and severe thrombotic thrombocytopenia [5]. In May 2021, Muir et al. described a case of a severe thrombocytopenia and disseminated intravascular coagulation that resembled autoimmune heparin-induced thrombocytopenia in a patient who had received the AD26.COV2.S vaccine [6]. Here, we describe a case of extensive thrombosis and adrenal haemorrhage associated with severe thrombocytopenia and disseminated intravascular coagulation that resembled the case of autoimmune heparin-induced thrombocytopenia in a patient who had received the Ad26.COV2.S vaccine (Johnson & Johnson, New Brunswick, NJ, USA), a recombinant adenovirus serotype 26 vector encoding the SARS-CoV-2 spike glycoprotein.

## 2. Case

A 39-year-old man was referred to our intensive care ward at the University Hospital Regensburg in Germany after he presented to the emergency department of the referring hospital two days before with severe pain in the left thorax and upper abdomen. Ten days earlier, he received a single dose of the SARS-CoV-2 vaccine Ad26.COV2.S (Johnson & Johnson, New Brunswick, NJ, USA). The patient was healthy until day 8 post vaccination, had no medical history, did not take any medication, and had no family history of venous thromboembolism. However, the patient smoked approximately 10 cigarettes a day for approximately 20 years.

Laboratory chemical analyses and a computed tomography (CT) of the thorax and abdomen were carried out at the referring hospital. Upon admission, 124/nL (norm 163–337/nL) platelets were documented. In a CT-scan on the day of admission, there was no evidence of pulmonary embolism or a thoracic or abdominal haemorrhage. Three days after initial presentation, the serum platelet level continued to decrease. On the day of transfer to our university hospital, 20/nL (norm 163–337/nL) platelets were registered.

At clinical examination on day 1 on ICU, the patient was oriented to person, place, time, and situation. A body temperature of 39.4 °C was documented, a sinus rhythm with a heart rate of 120 beats per minute was initially evident, blood pressure was 150/100 mmHg, and peripheral oxygen saturation was 96% while breathing room air. The body mass index was 26.5 kg/m^2^. The patient did not report headache or dyspnoea and did not show any signs of a deep vein thrombosis. He did not have any petechial haemorrhages. In addition to severe abdominal pain, physical examination did not reveal any pathological findings. SARS-CoV-2-RNA was not detected in a reverse transcription polymerase chain reaction assay of a sample obtained with a nasopharyngeal swab.

Blood specimens were obtained and sent to the respective laboratories for microbial and laboratory analysis located at our clinic. On day 3, laboratory parameters showed a thrombocytopenia of 6/nL (norm 163–337/nL) (Figure 1). Serum D-dimer concentration was elevated to 34 mg/dL (normal < 0.5 mg/dL) whereas fibrinogen concentration was 393 mg/dL and quite normal (normal 210–400 mg/dL). The activated partial thromboplastin time (aPTT) was 41 s (normal 25.9–36.6 s) and was slightly prolonged. The antibodies characteristic of heparin-induced thrombocytopenia (HIT) Type II were not identified. A thrombophilia screening, tests for antiphospholipid antibodies (cardiolipin, β2 glycoprotein, and lupus), and tests for paroxysmal nocturnal haemoglobinuria remained without pathological findings.

No schistocytes were detected and the activity of a disintegrin and metalloproteinase with a thrombospondin type 1 motif, member 13 (ADAMTS13) was in the normal range. Infections with human immunodeficiency virus, hepatitis B and C virus, hantavirus, rubellavirus, parvovirus B19, cytomegalovirus, and the Epstein–Barr virus were excluded.

Further diagnostics via the compression ultrasound of both legs showed a femoropopliteal and calf deep vein thrombosis in the right leg. Accordingly, suspected diagnosis was VITT. High-dose intravenous immunoglobulin (1 g/kg body weight) was administered and repeated 24 h later. On the same day, anticoagulant therapy with intravenous argatroban, a direct thrombin inhibitor, was initiated at a dose of 2 µg/kg body weight per min. The dose was increased to achieve an aPTT of 60–80 s. This treatment caused an increase in the platelet count from 6/nL to 77/nL within 7 days (Figure 1). Anti-platelet factor 4 (PF4)/heparin antibodies of the IgG class were detected with a high titre in the patient’s serum. This laboratory finding is consistent with COVID-19 vaccine-induced VITT.

As the condition of the patient rapidly improved, he was transferred from the intensive care unit to a normal ward on day 6 after admission. Two days later, the patient again reported severe pain in the left thorax and abdomen. CT pulmonary angiography showed the bilateral central emboli as well as emboli in the segmental arteries of every pulmonary lobe (Figure 2). Several consolidations were consistent with pneumonia because of pulmonary infarction. Magnetic resonance imaging of the abdomen showed the new onset of enlarged adrenal glands on both sides as a result of an acute bleeding event (Figure 3).

A transthoracic echocardiography showed mild right ventricular dilatation and troponin I levels were slightly elevated. Thus, the patient was re-transferred to the intensive care unit. Upon admission to the ICU, the patient was tachycardic and had a heart rate of 130/min, the mean arterial pressure was 75 mmHg, and peripheral oxygen saturation was 85% under room air. Therapeutic anticoagulation with argatroban was continued, analgesia was intensified, and oxygen was delivered by nasal mask.

Diagnosis was pulmonary embolism and primary adrenal insufficiency as a result of adrenal haemorrhage and consequently a therapy with hydrocortisone was initiated. Cortisol was measured in the serum and basal cortisol level of 2.5 µg/dL confirmed adrenal insufficiency, which is characterised by values below 5 µg/dL.

During the following days, the health status of the patient improved and five days later he could be discharged from the ICU. Intravenous anticoagulation via argatroban was changed to oral rivaroxaban. For follow-up, abdominal magnetic resonance imaging (MRI) was performed on day 10 (one day before discharge). In this imaging, analogous to the CT on day 9, adrenal lesions appeared marginally hyperintense and centrally hypointense in T2 weighting, consistent with haemorrhage (Figure 3). A short Synacthen test was performed 24 h after the last hydrocortisone dose. Cortisol levels were 1.8 µg/dL, 1.9 µg/dL, and 2.0 µg/dL (aim > 22 µg/dL) at 0, 30, and 60 min after ACTH (adrenocorticotropic hormone) administration, respectively, confirming the diagnosis of primary adrenocortical insufficiency. Therapy with 30 mg hydrocortisone and 0.05 mg fludrocortisone per day was continued. The patient was discharged from the hospital on day 22 with normal platelet counts and in good health.

### Published Cases of Adrenal Haemorrhage after Vaccination—Review of the Literature

To the best of our knowledge, five cases with adrenal haemorrhage after vaccination with a recombinant adenovirus serotype 26 vector encoding the SARS-CoV-2 spike glycoprotein have been reported to date. One of these cases was a 55-year-old woman, who came to the emergency department of Chelsea and Westminster Hospital NHS Foundation Trust in London [7]. She had received her first dose of the AstraZeneca COVID-19 vaccine 8 days before. The patient presented with vomiting and left iliac fossa pain. Later, a diagnosis of isolated thrombocytopenia was made. Platelet count decreased to 13 × 10^9^/L. The patient was transferred to the intensive care unit and treated with low molecular weight heparin (enoxaparin). Thromboembolism of the lungs, left basilic vein, and left renal vein were ascertained by imaging. Although the CT at admission did not reveal any abnormalities regarding the adrenal glands, a few days after admission, the left gland was enlarged. Hyperplasia of the right adrenal gland was also detected. The suspected diagnosis of adrenal insufficiency was confirmed by a short Synacthen test, which revealed cortisol levels of up to 95 nmol/L at 30 min. The patient was treated with hydrocortisone, and after recovery from severe illness, she received oral anticoagulation (apixaban).

A further case report was published from the authors working at the Hospital Universitario HM Montepríncipe in Madrid [8]. A 47-year-old man was admitted to the hospital for pulmonary embolism 10 days after having received the AstraZeneca COVID-19 vaccine. The patient did not suffer from any severe diseases before. High D-dimer levels, low platelet count, and PF4 antibody detection led to the diagnosis of VITT. Initial therapy with low-molecular weight heparin was replaced by subcutaneous fondaparinux and intravenous immunoglobulins. The patient reported abdominal pain 10 days later. Bilateral adrenal haemorrhage was diagnosed by MRI and laboratory tests. At the time of admission, this patient also had cerebral venous thrombosis which resolved during the first 10 days of hospitalisation.

A third and very severe case was reported from Naples [9]. A 54-year-old woman was admitted to the emergency room and examination of the patient revealed disseminated intravascular coagulation and thrombosis at multiple locations including the temporal lobes and the right frontal lobe. She had been vaccinated with the AstraZeneca vaccine 12 days before. A medical history of Meniere’s disease had been documented. An abdominal CT revealed adrenal haemorrhage. The patient died five days after the admission to the hospital with the diagnosis of a disseminated intravascular coagulation.

A Danish group described the case of a 60-year-old woman who was admitted to the hospital 7 days post-vaccination with the AstraZeneca vaccine [10]. She complained about abdominal pain, and a CT scan showed bilateral adrenal haemorrhages. Platelet counts dropped during hospitalisation and platelet concentrates were infused. Later on, anti-PF4 antibodies were detected in the patients’ blood samples. On the second day, the patient suffered a severe ischemic stroke and treatment with hydrocortisone and cefuroxime was initiated. The patient died six days after hospital admission. This patient had suffered from Hashimoto thyroiditis and hypertension and had otherwise been healthy.

A further case was described from the University Hospital of Wales [11]. A 38-year-old male patient with no medical history of severe diseases came to the emergency unit and complained about severe abdominal pain. At the day of admission, an increased white blood cell count and lactate level and mild thrombocytopenia was noticed. CT of the abdomen diagnosed abdominal haemorrhage and random cortisol level was low, requiring therapy with hydrocortisone. The platelet count markedly decreased during the following days and PF4 antibody screening was positive. Therapy with intravenous immunoglobulins, methylprednisolone, and argatroban was initiated, but adrenal haemorrhage progressed and thrombosis was found at different sites including the pulmonary arteries. Thus, plasma exchange was initiated, and under this intervention, platelet counts improved. The patient left the hospital with the diagnosis of primary adrenal insufficiency.

## 3. Discussion

The first cases of VITT after vaccination with the AstraZeneca vaccine (ChAdOx1) were described in February 2021 and were reported two months later for patients having received a Johnson & Johnson vaccination [12,13].

Regarding the mechanism by which immune thrombotic thrombocytopenia (ITT) is induced by an adenoviral vector-based COVID-19 vaccine, the main hypothesis relates to the reaction between the cationic PF4 and the anionic-free DNA contained in the recombinant adenovirus vaccine [14]. In a mouse model from 2013, DNA and RNA were shown to form multimolecular complexes with PF4 and expose the epitope to which anti-PF4/heparin antibodies bind, eliciting an immune response similar to HIT [15]. At present, there are no known differences in the pathophysiology or risk factors of VITT with the AstraZeneca vaccine (ChAdOx1) compared with the Johnson & Johnson vaccine (AD26.COV2.S).

Pavord et al. described the case definition criteria for VITT according to an expert haematology panel.

Four groups were classified according to the likelihood of VITT [16]. In our case, all five diagnostic criteria were fulfilled. The onset of symptoms 5–30 days after vaccination against SARS-CoV2, the presence of thrombosis, D-dimer level > 4000 FEU, thrombocytopenia and positive anti-PF4 antibodies on ELISA.

VITT is initiated by PF4 antibodies, which activate platelets and coagulation, leading to the formation of thrombi in vessels [12]. Antibodies against the SARS-CoV-2 spike protein did, however, not cross-react with PF4. Inversely, purified anti-PF4 antibodies from VITT patients did not cross-react with the recombinant SARS-CoV-2 spike protein. Anti-PF4 antibodies were detected in 8.6% of COVID-19 patients but could not activate platelets [14]. Therefore, it is unlikely that VITT was induced by the immune response against the viral spike protein [14]. The underlying pathways by which SARS-CoV-2 vaccination with adenoviral vectors stimulates anti-PF4 antibody production need further examination.

Adrenal haemorrhage and subsequent adrenal insufficiency occur when thromboembolic events affect the adrenal veins. The 39-year-old patient presented here developed VITT with consecutive venous thrombosis, central bilateral pulmonary embolism, and adrenal haemorrhage after vaccination with Ad26.COV2.S. VITT with adrenal insufficiency to our knowledge has not previously been described after immunisation with the Johnson & Johnson vaccine, and all known cases were associated with the AstraZeneca vaccine. VITT is a very rare adverse event of these vaccines but may become life-threatening, especially if left untreated [17]. Treatment recommendation for VITT includes non-heparin anticoagulants, immunoglobulins, and plasma exchange whereas, the transfusion of platelets should be avoided [17].

The patient described here was treated with intravenous immunoglobulin, argatroban, and because of adrenal insufficiency, hydrocortisone. There is growing evidence for the effectiveness of high-dose intravenous immune globulin (IVIG) plus anticoagulation in the treatment of VITT [18,19,20].

Encouragingly, there was rapid clinical and laboratory improvement after treatment initiation. However, despite the early initiation of therapeutic anticoagulation, the patient developed severe pulmonary embolism after 8 days of the continuous application of argatroban. The choice of anticoagulant in VITT is based on its pathophysiologic similarities with autoimmune type 2 HIT (heparin-induced thrombocytopenia). Our experience relating to the presented patient case suggests that VITT may cause similar symptoms in comparison to autoimmune HIT, in which bilateral adrenal haemorrhage has been previously reported [21,22]. In our case, adrenal haemorrhage and pulmonary embolism were diagnosed during ongoing treatment with argatroban. Therefore, we consider it likely that the triggering of the thromboembolic and bleeding event was caused by the vaccine. Additionally, our case might indicate that thromboembolism in VITT can be refractory to anticoagulation and requires increased awareness during clinical care despite the initial improvement of the condition.

Concomitant with pulmonary embolism, the patient developed bilateral adrenal haemorrhage, for which there was no evidence at the initial CT scan, and which led to primary adrenocortical insufficiency. A few cases of vaccine-induced thrombocytopenia with bilateral adrenal haemorrhage after having received the adenoviral vector-based vaccine ChAdOx1 have been reported [6]. Based on our case, we conclude that vaccine-induced adrenocortical haemorrhage is a rare but significant adverse event of adenoviral vector-based COVID-19 vaccines.

Among the adverse events of the COVID-19 vaccine of Johnson & Johnson reported to the Vaccine Adverse Events Reporting System (VAERS), 97% were classified as non-serious and 3% as serious, including three female cases of thrombosis in large arteries or veins accompanied by thrombocytopenia during the second week after vaccination [23]. In addition, a case series of 12 US patients with cerebral venous sinus thrombosis and thrombocytopenia following the use of Ad26.COV2.S vaccine under Emergency Use Authorization were reported to the VAERS [24].

COVID-19 vaccine-associated cases of thrombocytopenia in Europe share laboratory and clinical characteristics with the syndrome of autoimmune heparin-induced thrombocytopenia (HIT) [12,25,26]. The presence of antibodies may constitute a link between the immune response to the vaccine and the clotting syndrome. VITT is characterised by exposure to one of the aforementioned vaccines 4–30 days prior to presentation, followed by thrombosis, mild-to-severe thrombocytopenia, and a positive PF4-heparin enzyme-linked immunosorbent assay (ELISA). Thrombosis involves atypical locations, including cerebral vein and splanchnic vein thrombosis [27].

The time course of the reported case and the occurrence of heparin-independent PF4 antibodies is in line with the published literature. Mortality from VITT was reported to be approximately 30–60% [12,25,26]. With an increasing number of vaccinated people, more and more adverse effects are being reported, including adverse events which have not been previously described. Clinicians must be aware of these rare cases, which possibly present with unspecific symptoms, and if not treated early, may be in a life-threatening situation.

## 4. Conclusions

To our knowledge, this is the first case of simultaneously diagnosed bilateral pulmonary embolism and adrenocortical haemorrhage with consecutive adrenocortical insufficiency after vaccination with Ad26.COV2.S. While VITT is very rare, its potentially life-threatening course requires early diagnosis and close monitoring for complications. At present, there are no known differences in the pathophysiology or risk factors of VITT with the AstraZeneca vaccine (ChAdOx1) compared with the Johnson & Johnson vaccine (AD26.COV2.S).

## Figures and Tables

**Figure 1 vaccines-10-00595-f001:**
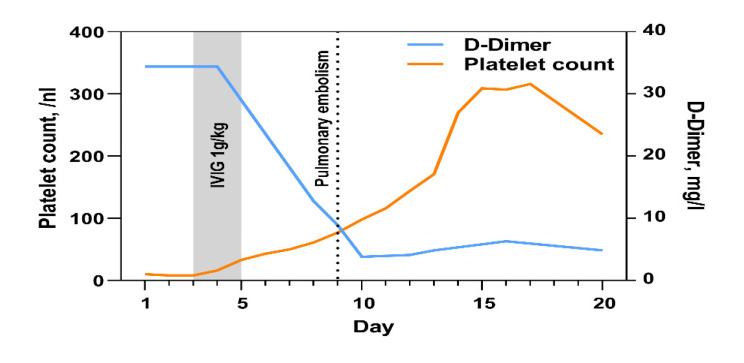
Platelet count and D-dimers during the first 20 days of hospitalisation.

**Figure 2 vaccines-10-00595-f002:**
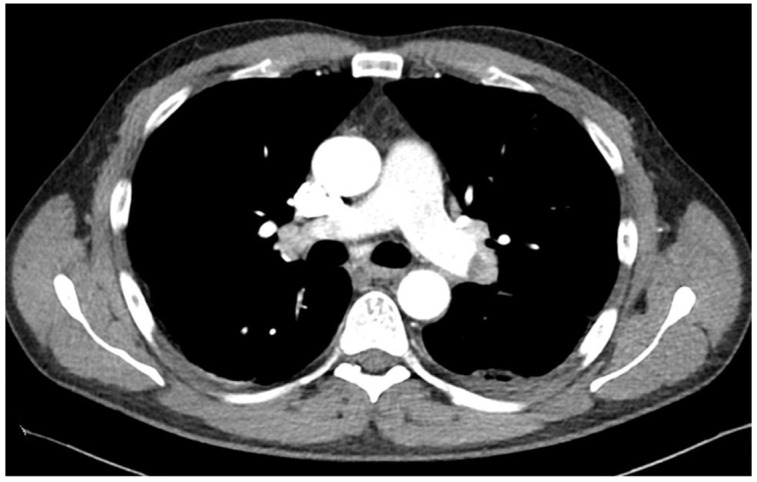
Thoracic computed tomography (CT) at day 9 after admission revealed a bilateral central pulmonary embolism.

**Figure 3 vaccines-10-00595-f003:**
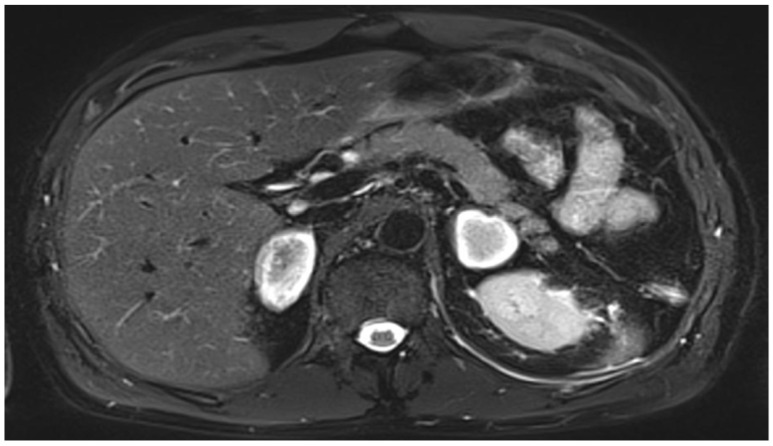
Abdominal magnetic resonance imaging (MRI) at day 21 after admission revealed a bilateral adrenal haemorrhage.

## Data Availability

Not applicable.

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
