# Peer review of "SARS-CoV-2 Vaccine-Induced Immune Thrombotic Thrombocytopenia with Venous Thrombosis, Pulmonary Embolism, and Adrenal Haemorrhage: A Case Report with Literature Review"

_vaccines, 2022, doi:10.3390/vaccines10040595_

Round 1

Reviewer 1 Report

In their manuscript entitled "SARS-CoV-2 vaccine-induced immune thrombotic thrombocytopenia with venous thrombosis, pulmonary embolism, and ad-3 renal haemorrhage: a case report with literature review", the authors reported a case of patient with VITT after having received the vaccine Ad26.COV2.S.  The clinical observation and treatment are well described. This report together with previous studies confirm the link between adenovirus vectored COVID vaccine and VITT.

I`d like to suggest  the authors to discuss a little bit about the potential mechanism, especially regarding the role of the adenovirus vector in VITT.

Author Response

Dear Reviewer,

thank you very much for your comments. I have integrated the pathophysiological considerations into the article.

Sincerely,

Hauke Tews

Reviewer 2 Report

The authors report the first case of simultaneously diagnosed bilateral pulmonary embolism and adrenocortical haemorrhage after vaccination with Johnson & Johnson Ad26.COV2.S, and then review 5 cases of adrenal haemorrhage caused by vaccination with an AstraZeneca adenovirus vector encoding the SARS-CoV-2 spike glycoprotein.

  1. By comparison this case related to Johnson & Johnson vaccine with the previously-reported cases of AZ vaccine, what is the main conclusion(s) from this study. This should be clearly presented in Discussion and add to Conclusion and Abstract.
  2. There are some minor editing works, for examples:

In line 31: "The COVID-19 pandemic so far has resulted in approximately 472 million confirmed cases and at least 6,1 million deaths. There are about 1,9 million cases every day…"

In line 50: "A 39-year-old man was referred to our intensive care ward at the University Hospital Regensburg, Regensburg. Germany after he had presented to the emergency department…"

In line 203: "Adrenal hemorrhage and as a consequence adrenal insufficiency develops when thromboembolic events affect the adrenal veins."

Author Response

Dear Reviewer,

thank you very much for your comments. I have edited the formal errors and highlighted your substantive emphasis and considerations. 

Sincerely,

Hauke Tews
